# Mycochemicals against Cancer Stem Cells

**DOI:** 10.3390/toxins15060360

**Published:** 2023-05-25

**Authors:** Massimo Tacchini, Gianni Sacchetti, Alessandra Guerrini, Guglielmo Paganetto

**Affiliations:** Department of Life Sciences and Biotechnology, University of Ferrara, 44121 Ferrara, Italy; massimo.tacchini@unife.it (M.T.); alessandra.guerrini@unife.it (A.G.); guglielmo.paganetto@unife.it (G.P.)

**Keywords:** mycochemicals, cancer stem cell, medicinal mushroom

## Abstract

Since ancient times, mushrooms have been considered valuable allies of human well-being both from a dietary and medicinal point of view. Their essential role in several traditional medicines is explained today by the discovery of the plethora of biomolecules that have shown proven efficacy for treating various diseases, including cancer. Numerous studies have already been conducted to explore the antitumoural properties of mushroom extracts against cancer. Still, very few have reported the anticancer properties of mushroom polysaccharides and mycochemicals against the specific population of cancer stem cells (CSCs). In this context, β-glucans are relevant in modulating immunological surveillance against this subpopulation of cancer cells within tumours. Small molecules, less studied despite their spread and assortment, could exhibit the same importance. In this review, we discuss several pieces of evidence of the association between β-glucans and small mycochemicals in modulating biological mechanisms which are proven to be involved with CSCs development. Experimental evidence and an in silico approach are evaluated with the hope of contributing to future strategies aimed at the direct study of the action of these mycochemicals on this subpopulation of cancer cells.

## 1. Introduction

Tumour masses exhibit significant cellular heterogeneity, containing various types of cells, including stroma cells, endothelial cells, and immune cells. Studies have revealed that numerous forms of cancer arise and persist through the actions of a limited number of cancer stem cells (CSCs) [1,2,3,4]. These CSCs generate most of the tumours through ongoing self-renewal and differentiation, which could be managed by the same signalling pathways observed in normal stem cells. In the past two decades, the power of CSCs in creating new tumours during experimental implantation in animals has confirmed the relevance of treatments with chemical compounds aimed at this specific cell type [5,6]. Targeting CSCs is crucial for effective cancer treatment, especially with the high relapse rates seen with current chemo and radiotherapy methods. Moreover, a strong relationship exists between CSCs and drug resistance in cancer patients. Screens for agents that specifically kill CSCs require a complex, laborious, and expensive effort due to the rarity of these cells within tumour cell populations and their relative instability in culture [7]. Natural molecules have the potential to target various cellular pathways that can help eliminate CSCs. These molecules may work by inhibiting self-renewal and stemness pathways, such as Notch, Hedgehog, and Wnt, blocking antiapoptotic signals and promoting apoptosis. They can also prevent drug efflux by blocking ATP-binding cassette transport proteins and reducing survival pathways such as phosphoinositide 3-kinase/Akt and extracellular signal-regulated kinases. While there have been several scientific reports on the effectiveness of botanicals against CSCs, the research is still limited, especially regarding the mushroom kingdom. Despite many scientific studies that have been undertaken on the use of mushroom extracts against cancer, few of them specifically address CSCs. Thus, this review aims to provide an overview of various mycochemicals derived from dietary and non-dietary mushrooms that could display the capacity to selectively interfere with signalling pathways involved in CSCs growth and control.

In this context, indirect immunomodulation could be mediated by macromolecules, like polysaccharides and polysaccharopeptides, or small molecules, which could interfere with the pathways involved in stemness.

In the bibliographic research for this review, we identified and utilised relevant keywords pertaining to the biological mechanisms underlying the growth and proliferation of cancer stem cells (CSCs). Each section of our review was titled using these keywords as a guide. Subsequently, we conducted a comprehensive investigation of various fungal species and their corresponding metabolites that have demonstrated significant activity within these pathways.

On the one hand, the relevance of using phytochemicals extracted from plants against cancer stem cells is widely documented in the literature by several reviews [8,9,10]. On the other, despite the growing increase of similar studies on extracts of fungal origin, our research has uncovered a significant shortage in the availability of an all-inclusive database containing the bioactivities of mycochemicals towards the pathways involved in the biology of CSCs. Our effort is oriented to contribute to overcoming this shortage.

Therefore, this review aims to provide a methodological and effective tool to score mushroom mycochemicals according to the CSCs pathways interference probability.

## 2. Indirect Immunomodulation on CSCs. β-Glucans and β-Glucans Derivatives

β-glucans are the best-known and most potent mushroom-derived macromolecules with anti-tumour and immunomodulating properties. These polysaccharides consist of a backbone of glucose residues linked by β-(1-3)-glycosidic bonds, often with attached side-chain glucose residues joined by β-(1-6) linkages [11] (Figure 1). β-glucans are the most relevant metabolite due to their broad spectrum of biological activity. Stimulation of both humoral and cellular immunity and inhibition of cancer are the well-studied effects of β-glucans. Recent research has revealed that β-glucans have not demonstrated any direct cytotoxic effects on a range of cell lines, including carcinoma, sarcoma, and blastoma, through in vitro testing. Furthermore, there is no indication that they directly activate apoptotic pathways or impact cancer cells’ telomerase and telomeric length. In contrast, β-glucans show potent anti-cancer activities as immune-stimulating agents [12,13], exhibiting inhibition of tumour growth, inhibition of vascular endothelial growth factors (VEGF) and metalloproteinases 2 and 9 (MMP-2 and MMP-9) modulation [14]. Evidence has indicated that β-glucans possess the potential to reduce the size of xenografted cancerous tumours and can be used in conjunction with other immune stimulants and chemotherapy drugs. A promising approach involves utilising the transporting properties of β-glucans to deliver nanoparticles containing chemotherapy agents directly to the cancer site, thus enhancing the overall efficacy of therapy [12].

Nevertheless, medicinal mushroom extracts exert a direct in vitro anti-tumour activity on cancer cells by inhibiting cancer cell adhesion, reducing integrin expression [15], and inhibiting tumour cell proliferation [16]. This evidence leads to the hypothesis of the presence of other molecules that may act in synergy with β-glucans to promote anti-cancer activity.

**Figure 1 toxins-15-00360-f001:**
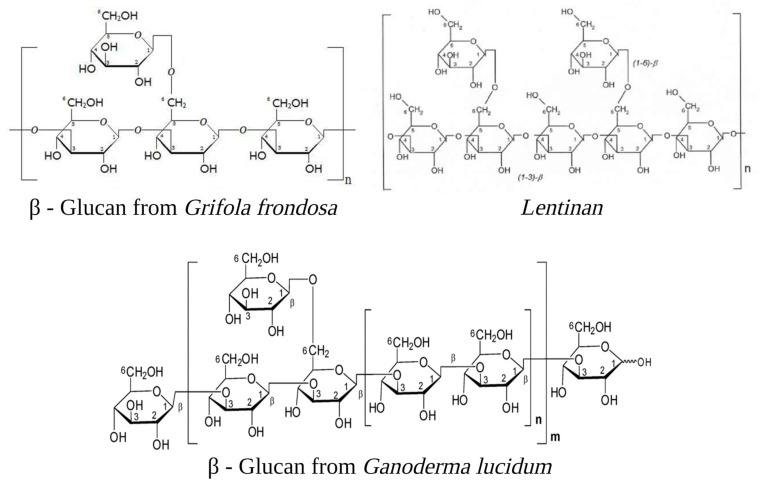
Examples of β-glucans are cited in the text: Grifolan (Molecular Weight 1200 kDa) [17], Lentinan (MW 500 kDa) [18], and GLP20 (MW 3.75 MDa) [19]. The diagram presented exclusively displays the fundamental building blocks forming the complex molecular structure. The image was created in BioRender.com (Agreement number: ZM254U57YL).

CSCs are resistant to chemotherapy, possibly because they possess unique glycosaminoglycans and proteoglycans that provide a niche for the preservation of cellular “stemness”, as shown by the observation that syndecan-1-deficient mice are largely resistant to mammary tumour formation [20]. The syndecan family of transmembrane proteoglycans is the primary cell surface heparan sulfate source in all cell types. The syndecan-1 protein functions as an integral membrane protein and participates in cell proliferation, cell migration, and cell–matrix interactions via its receptor for extracellular matrix proteins—these observations oriented research of anti-CSC towards the inclusion of proteoglycan fragments like polysaccharopeptides (PSPs).

PSPs comprising β-D-glucans are covalently linked through O-linked and/or N-linked glycosidic bonds to a small aspartate glutamate-rich polypeptide. PSPs are heterogeneous in size, with an average molecular weight of 100 kDa. Following its discovery and partial characterisation, PSP has garnered significant interest from the scientific community due to its numerous health benefits. The diverse biological effects span immunomodulation, suppression of cell proliferation, attenuation of cell cycle control, regulation of tumour-specific gene expression, induction of apoptosis in human leukaemia cells, mitigation of adverse effects of chemotherapy, and increased survival among cancer patients [21,22]. Luk et al. [23] demonstrated that the PSPs isolated from cultured mycelia of the Cov-1 strain of *Trametes versicolor* (L.) Pilàt target prostate CSCs in vitro and suppress tumour formation in vivo. Treatment of prostate cancer cell line PC-3 with PSP led to the downregulation of CSC markers (CD133 and CD44) in a time and dose-dependent manner. Furthermore, PSP pre-treatment significantly suppressed tumour initiation of PC-3 cells in immunodeficient mice, suggesting that PSPs suppress the tumourigenicity of the PC-3 cells. It is worth noting that the administration of PSPs through oral feeding to transgenic mice that naturally develop prostate tumours resulted in the complete suppression of prostate tumour growth. This finding suggests that PSPs may serve as a potent chemopreventive agent against prostate cancer, potentially by targeting the prostate cancer stem cell (CSC) population. The relevance of PSPs on prostate CSC control was also highlighted in two later studies reporting the control of CSCs and the gut microbiome by the PSP-derived mycelia of the medicinal mushroom *T. versicolor* strain [15] and the occurrence of potent anti-cancer synergistic activities through targeting of CSCs between PSP and gamma-tocotrienol [24].

Although these polysaccharides are typically considered indigestible carbohydrates, they can still undergo breakdown into smaller polymers via fermentation by intestinal microbes or within macrophages following internalisation. Research on this topic has yielded mixed findings. While larger β-glucans can independently trigger an immune response, partial degradation can also enhance their desired effects. However, unregulated hydrolysis may occur, mainly through endoglucanases, resulting in a range of compounds with uncertain activities and a loss of immunostimulatory properties [25]. The substitution of O-glycosidic linkages by thiobridges will positively increase the lifetime of glucans and, therefore, biological impact in vivo. Indeed, thioglycosides are known to be more beatable than their O-counterparts against acidic or enzymatic conditions, and this chemical modulation has only a minor effect on the tridimensional arrangement. Sylla et al. [25] reported the synthesis of small oligo-β-(1-3)-glucans characterised by thioglycosidic linkages and showed that the presence of sulfur atom(s) was not only crucial to prolong in vivo immunoactivity compared with native polysaccharides but significantly suppresses the proliferation of colon cancer stem-like cells from human colon adenocarcinoma.

### 2.1. Tumour-Associated Macrophages

Several studies have clarified the importance of tumour-associated macrophages (TAM) as major contributors in regulating self-renewal and anticancer drug responses of CSCs through distinct networks of cytokines, chemokines, and growth factors. In these processes, TAMs interact with and promote the oncogenic potential of cancer stem cells via the production of milk-fat globule epidermal growth factor–VIII (MFG-E8) and IL-6 through coordinated activation of the STAT3 and Sonic Hedgehog (SHH) pathways [26]. CSCs are the major subset promoting the production of MFG-E8 and IL-6 from macrophages, implying that mediators specifically regulated by CSC render macrophages able to facilitate the production of tumourigenic factors. MFG-E8 selectively modulate macrophage activation and polarisation in M2-tumour-associated macrophages, which promote tumour growth, progression, and metastasis [27].

Studies conducted in vivo have demonstrated that oral β-glucan can notably impede the growth of tumours associated with TAM phenotype conversion and promote effector T cell activation. Further, mice injected with particulate β-glucan-treated TAM along with tumour cells have shown significantly decreased tumour burden, with fewer blood vascular vessels compared with those who received TAM plus tumour cell injections. The same research has also indicated that β-glucans, through activating dectin-1, can convert M2 macrophages to an M1-like phenotype, facilitate the production of reactive oxygen species (ROS), activate NF-κB, and subsequently secrete proinflammatory cytokines [28].

### 2.2. Inhibitory Signalling Proteins PD-L1 and CTLA-4

Cancer cells express many inhibitory signalling proteins that cause immune cell dysfunction. Two of these inhibitory molecules are programmed cell death ligand 1 (PD-L1) and cytotoxic T-lymphocyte-associated protein 4 (CTLA-4). The activated T cells express programmed cell death receptor 1 (PD-1), which can be engaged to induce inhibitory signalling downstream of the T-cell antigen receptors (TCRs). This process effectively blocks the effector functions of the T cells [29,30]. The latter, when bound to another protein called B7, helps keep T cells from killing other cells, including cancer cells. Fungal-derived β-glucans have also been shown to interfere with these pathways: lentinan, a β-glucan extracted from the basidiomycete *Lentinula edodes* (Berk.) Pegler, significantly reduced PD-L1 expression at the transcriptional levels in gastric cancer [31]; additionally, the extract from the sporoderm-breaking spore of *Ganoderma lucidum* (Curtis) P. Karst. downregulated the two immune checkpoints, programmed cell death PD-1 in the spleen, and CTLA-4 in the breast cancer cells [32] (Figure 2). This evidence suggests that the extract could effectively restore the T cell activity by recovering the exhaustion status via suppressing the co-inhibitory checkpoints.

Research has revealed a notable association between the manifestation of PD-L1 and CTLA-4 and the presence of cancer stem cell (CSC)-like properties, including elevated levels of CD44 and/or CD133 expression on neoplastic cells [7,33]. PD-L1 monoclonal antibodies inhibitors, hence, have been considered a therapeutical approach. Despite their success in clinical trials, they still show problems, including higher production costs, lower oral bioavailability, poor tumour penetration, and immune-related adverse events. In comparison to peptides, small molecules offer advantages in terms of their oral and plasma stability, as well as their bioavailability. However, due to the complexity and plasticity of the PD-L1 surface, the development of effective small-molecule inhibitors targeting PD-L1 poses a significant challenge [34]. In this context, β-glucans possess the advantages of low toxicity, and long-term consumption appears to have no side effects. The results of this study indicate that consumption of the aforementioned compounds has the potential to serve as a safe and efficient treatment for cancer, exhibiting pharmacokinetic properties akin to those of small molecules [35].

### 2.3. Toll-Like Receptors (TLRs)

Toll-like receptors (TLRs) are pattern recognition receptors that play a key role in the activation of innate immunity due to their ability to sense pathogen-associated molecular patterns (PAMPs) and danger-associated molecular patterns (DAMPs) [36]. The role of their expression in cancer cells is unclear, but in some cases, it seems associated with cancer stem cell behaviour [37]. The activation of TLRs in lung and breast cancer leads to an increased expression of β-catenin and nuclear factor kappa-light-chain-enhancer of activated B cells (NF-κB) signalling pathways causing a phenotypic switch toward a CSC phenotype, increasing their expansion, invasion, and metastatic potential [38,39,40]. Increased stem-like properties were also associated with high TLR4 expression in mice models of hepatocellular carcinoma (HCC). It significantly enhances cell invasion and migration [41], and it promotes the formation of stem-like cancer cells [42]. Furthermore, experimental data from Li et al. [43] showed that downregulation of TLR4 expression significantly suppressed cell proliferation, cell migration, cell invasion, induced tumour apoptosis in vitro, and suppressed tumour growth in vivo in lung cancer cells. β-glucans from medicinal mushrooms are one of the major PAMPs and are involved in innate immune cell initiation. The interaction between β-glucan and its receptor molecules, such as dectin-1, CR-3, or TLRs, which are expressed by immune cells, has the potential to influence the immune response to cancer cells [44].

For instance, the polysaccharide fraction of *Agaricus blazei* Murrill mycelium, obtained by water extraction, seems to induce the expression of IL-12 in human peripheral mononuclear cells (PBMC) transduced by TLRs. This cytokine, a critical regulator of cellular immune responses, showed anti-cancer activity, inducing a robust immune response against cancer cells and acting as an anti-angiogenic agent [45]. In addition, research has demonstrated that interleukin-12 (IL-12) can mitigate the formation of tumourspheres by CSCs, leading to a decrease in proliferation and an increase in apoptosis levels in vitro. Moreover, IL-12 has been found to enhance the expression of a differentiation marker, indicating its capacity to activate signals for cell differentiation [46].

The alkaline polysaccharide fraction isolated from the fruiting body of *Pleurotus ostreatus* (Jacq.) P. Kumm. act as a ligand for several receptors on the surface of macrophages, including TLR-4, showing a significant increase in the level of TNF-α and an enhancement of macrophages’ phagocytic capability [47]. Water-soluble polysaccharides from *Pleurotus eryngii* (DC.) Quél. fruiting body and a polypeptides fraction from the mycelium extract of the same mushroom were found to enhance the expression of TLR2 and TLR4, also leading to a dose-dependent increase in the release of TNF-α [48,49]. Research has revealed that the inhibition of TLR-4 can effectively inhibit the production of IL-12 p40 and IL-10 induced by purified *G. lucidum* glucans (PS-G). This highlights the crucial role of TLR-4 signalling in the maturation of dendritic cells induced by glucans [50].

The immunomodulating properties of a 450 Kd β-glucan from *Pleurotus citrinopileatus* Singer (PCPS) have been observed using primary human monocyte-derived, dendritic cells (DCs), which express various PRRs, such as TLR receptors and C-type lectins. The modulatory effect of mushroom polysaccharides in human dendritic cells was found to be significantly augmented through the activation of TLR4 and further enhanced synergistically upon stimulation of TLR2. These findings indicate that the interaction between PCPS and these TLRs plays a crucial role in the observed modulation of dendritic cells [51].

Polysaccharides extracted from *L. edodes* have been used as an adjuvant of chemotherapy, and one of its lately discovered polysaccharides, called MPSSS, inhibited the VEGF-C secretion of cancer-associated fibroblasts (CAFs) via the TLR4/JNK pathway, reducing lymphangiogenesis and lymphatic metastasis of colorectal cancer in a mouse model [52]. Batbayar et al. [53] demonstrate that β-glucans extracted from *G. lucidum* bind directly to the dectin-1 and TLR-2/6 receptors on the macrophage cell surface, producing the secretion of pro-inflammatory cytokines and induction of cellular TLRs.

### 2.4. Natural Cytotoxicity Receptors Ligands (NCR)

Natural killer cells (NK cells) have been described to efficiently recognise and kill in vitro CSCs isolated from colorectal cancer (CRC), melanoma, and glioblastoma [54,55,56,57]. NK cell function is triggered by the loss of the inhibitory receptor-mediated signalling when tumour cells downregulate the expression of MHC (major histocompatibility complex) class I. The simultaneous engagement of activating receptors, such as NKG2D, DNAM-1, and natural cytotoxicity receptors NKp30, NKp44, and NKp46 is required for the robust activation of NK cells and the NK cell-mediated killing of transformed cancer cells. Moreover, the efficiency of NK cell-mediated lysis of CSCs is also dependent on the expression of the natural cytotoxicity receptors ligands [58] and from the suboptimal or negative expression of HLA class I molecules on the surface of CSCs [54,55,56,59]. Interestingly, Tallerico et al. [60] found that CSCs are susceptible to NK cells but not their differentiated counterpart of CRC. Studies have shown that the quantity of ligands of activatory NK receptors on CSCs is a crucial factor in promoting effective innate immune responses, as demonstrated in glioblastoma multiforme and melanoma cases [54,55,60]. In fact, in patients with acute myeloid leukaemia (AML), the suboptimal expression of NKG2D ligands has been described as an escape mechanism of tumour cells from NK cell recognition [61], confirming that these molecules can affect the susceptibility of cancer cells to innate responses. Therefore, the expression patterns of NKG2D ligands in tumour cells, including CSCs, could serve as a predictive marker for determining the appropriate type of immunotherapy intervention. El-Deeb et al. [62] investigated the possible immunomodulatory effects of polysaccharides extracted from fruiting bodies of *P. ostreatus* on NK cells against different cancer cells. In this study, the activation effect of various fractions was evaluated on natural killer (NK) cells against three distinct cancer cell lines. The experiment was conducted under three different activation and co-culture conditions in the presence or absence of human recombinant interleukin-2 (IL2). The possible modes of action of mushroom polysaccharides against cancer cells were assessed at cellular and molecular levels. The study revealed that *P. ostreatus* polysaccharides had a cytotoxic effect on NK cells against lung and breast cancer cells, with the most significant effect observed against breast cancer cells (81.2%). It was observed that the activation of NK cells for cytokine secretion was associated with the upregulation of KIR2DL genes, while the cytotoxic activation effect of NK cells against cancer cells was linked with NKG2D upregulation and the induction of IFNγ and NO production.

Furthermore, the effects of polysaccharides from *Grifola frondosa* (Dicks.) Gray, *L. edodes,* and *G. lucidum* were investigated on primary human NK cells under normal or simulated microgravity (SMG) conditions [63]. Authors demonstrated that polysaccharides markedly promoted the cytotoxicity of NK cells by enhancing IFN-γ and perforin secretion and increasing the expression of the activating receptor NKp30 under normal conditions. In SMG conditions, instead, polysaccharides can enhance NK cell function by restoring the expression of the activating receptor NKG2D and by reducing early apoptosis and late apoptosis/necrosis.

### 2.5. Transmembrane Protein Receptor Dectin-1 Cluster

The dectin-1 cluster is a group of type II transmembrane protein receptors expressed on cells responsible for innate immune response, such as macrophages, dendritic cells, and neutrophils. It is a subgroup of the C-type lectin receptors (CLRs) group, which plays diverse functions ranging from embryonic development to immune function [64]. This cluster includes several receptors such as dectin-1, CLEC-2, CLEC-9A, MelLec, LOX-1, CLEC12B, CLEC12A, and MICL. The latter two are considered promising therapeutic and diagnostic targets in acute myeloid leukaemia (AML) and myelodysplastic syndromes (MDS); in particular, the 1C-type lectin domain family 12 member A (CLEC12A) is an inhibitory receptor identified as a specific marker for CSC in acute myeloid leukaemia [64,65]. The myeloid inhibitory C-type lectin-like protein, or MICL, is predominantly expressed by myeloid cells. Its cytoplasmic tail contains an immunoreceptor tyrosine-based inhibition motif (ITIM) that can recruit Src homology region 2 domain-containing phosphatases (SHP-1 and SHP-2). Through this pathway, MICL is capable of negatively regulating inflammatory cellular responses. Additionally, MICL is highly expressed in leukaemia stem cells but not in normal hematopoietic stem cells [66].

Within the receptor cluster, there exists a notable member known as dectin-1. This receptor, resembling that of a natural killer cell, functions as a transmembrane pattern recognition receptor. Its primary function is to bind β-glucan carbohydrates, thereby initiating and regulating the immunological response. This response includes phagocytosis and the production of proinflammatory factors. These immunological actions eliminate infectious agents and bolster the overall immune response of the organism [67]. The entire signalling pathway downstream to dectin-1 activation has not yet been fully mapped out, but several signalling molecules have been reported to be involved. They are NF-κB (through Syk-mediate pathway), signalling adaptor protein (CARD9), and nuclear factor of activated T cells [68,69]. The SIGN-related 1 (SIGNR1) protein, which is a dendritic cell-specific ICAM-3-grabbing non-integrin homolog, serves as a significant mannose receptor on macrophages, working in concert with dectin-1 to facilitate non-opsonic recognition of β-glucans for phagocytosis [70]. Moreover, the dectin-1 cytoplasmic tail harbours an immunoreceptor tyrosine-based activation motif (ITAM) that collaborates with TLR 2 and TLR6 to signal through the tyrosine kinase [71].

Wang et al. [72] demonstrated how a water-soluble homogeneous polysaccharide isolated from the fruit bodies of *G. frondosa* (GFPBW2) might be a potential ligand of dectin-1 and could also work as a macrophage activator by triggering cytokine secretion. These affirmations were based on the fact that GFPBW2 could bind dendritic cell-associated C-type lectin-1 (dectin-1) with an affinity constant (Kd) value of 1.08 × 10^−7^ M while it activated Syk and enhanced TNF-α production in RAW264.7. Furthermore, Syk, NF-κB signalling, and cytokine release in macrophages induced by GFPBW2 were significantly inhibited by a specific dectin-1 blocking reagent, laminarin. The binding between dectin-1 and β-glucans has also been demonstrated to stimulate the phenotypic and functional maturation of dendritic cells and induce the differentiation of monocytes towards macrophages with a reduced proinflammatory capacity [44].

### 2.6. Signal Transducer and Activator of Transcription (STAT) Family

The STAT protein family comprises latent cytoplasmic transcription factors that respond to cytokines and growth factors. Upon activation, these STATs move to the nucleus and bind to specific promoter elements of target genes, thereby regulating their transcription [73]. The STAT family is involved with many essential functions in the biology of normal cells and transformed cells. In particular, STAT3 was reported to have a crucial role in maintaining the expression of genes important for stem cell phenotype and used as markers of CSCs. The STAT3 pathway is preferentially active in subpopulations of cells enriched for CSC markers, and its inhibition decreases cell viability and tumoursphere formation. Moreover, STAT3 activation, triggered by IL-6 or TGF-β/LIF system, plays a role in EMT induction in different types of tumours [74]. For example, the activation of STAT3 signals maintains the self-renewal and tumourigenic potential of glioblastoma stem cell-like tumour cells (GSC). STAT3 is a critical signalling node in GSC maintenance, and its inhibition can suppress the growth of GSC-derived intracranial tumours [75]. STAT3 also plays a key role in breast cancer by acting as a transcriptional activator, regulating several target oncogenes and affecting breast cancer progression, proliferation, apoptosis, metastasis, and chemoresistance [76].

Rios-Fuller et al. [77] demonstrated on one side that a commercial *G. lucidum* extract (consisting of 13.5% polysaccharides, 6% triterpenes, and 1% cracked spores) reduced STAT3 gene mRNA abundance by greater than 50%, and by 35% in two triple-negative breast cancer cell lines (respectively in SUM-149 and MDA-MB-231) in vitro. On the other side, the same extract could significantly suppress multiple properties of CSCs in vivo: decreasing the ALDH1 population and the CD44^+^/CD24^−^ stem-like population, inducing mammosphere deformation and decreasing circularity, and blocking self-renewal transcription factors expression through inhibition of STAT3 signalling.

## 3. Indirect Immunomodulation on CSCs. Mushrooms Small Molecules

In the previous section, we deal with the interaction between some immune system macromolecules and fragments of macromolecules derived from the mushroom. Enriched fractions of polysaccharides and polysaccharopeptides represent a promising avenue for chemoprevention against cancer stem cells. However, there are unique advantages to modulating the immune system via intervention of small molecule (Figure 3). These include oral bioavailability, the ability to penetrate physiological barriers and achieve exposure within the tumour microenvironment, and well-understood formulation and dosing options that can mitigate pharmacokinetic and/or pharmacodynamic challenges while enabling titration of drug exposure. Unlike the extensive literature on the immune interaction of specific fractions or macromolecules extracted from mushrooms, single-molecule studies are scarce. In silico evaluations oriented to guide future experimental approaches could be effective tools to overcome this shortage. Maruca et al. [78] published a study reporting possible interactions occurring among bioactive natural fungal-extracted small molecules and several protein targets of therapeutic interest. A chemical database of compounds extracted from both edible and non-edible mushrooms was created. This database was virtually screened by in silico docking experiments against 43 macromolecular targets downloaded from the Protein Data Bank website and ranked according to the protein’s free energy interactions. This study has not been specifically geared toward cancer stem cells. However, it can be a valuable tool for setting interaction parameters between small molecules and protein receptors involved in cancer stem cell pathways. The cellular pathway studied by Maruca et al. [78] can be compared with relevant references in respect of the inflammatory response associated with CSCs because the inflammatory microenvironment is an essential component of the tumour microenvironment.

The CSCs niche homeostasis is regulated by a complex system of molecular signals, which also include the COX-derived arachidonic acid metabolites [79,80]. In greater detail, research has shown that COX-2 is increased in CSCs isolated from various tumour histotypes, including breast, colon, and bone tumours. Additionally, COX-2 is co-expressed with molecular markers of stemness and has been linked to the promotion of CSC growth in in vitro experimental systems [81]. This is not surprising considering the relevance of the COX-2-PGE2 system in the stem cell biology of normal tissues [82], as well as the well-known relevance of COX-2 in cancer. Some experimental evidence of COX-1 involvement in the biology of CSCs also exists. In the azoxymethane murine colon cancer model (AOM), the early molecular response of intestinal stem cells to genotoxic insult is driven by COX-1-PGE2 signalling and results in increased stem cell survival [83]. Breast CSCs obtained from primary cultures of spontaneous tumours in HER2/Neu transgenic mice exhibit upregulation of COX-1 and COX-2 genes [84]. Moreover, both COX isoforms belong to an eight-gene signature that correlates with breast cancer patient survival, thus suggesting a role of both isoforms in breast cancers with HER2 over-expression [85]. In recent times, research has revealed that the emergence of drug resistance in cancer treatment is also attributed to the activation of mesenchymal stem cells (MSCs), which can modulate the response to chemotherapy through various means [86]. A peculiar occurrence after cisplatin treatment is the secretion of specific polyunsaturated fatty acids that can confer resistance to cancer cells against different anticancer drugs. Interestingly, the central enzymes involved in the synthesis of MSC-derived chemoprotective factors are COX-1 and thromboxane synthase [87], thus suggesting that enzyme inhibition could restore cancer cell sensitivity.

Natural molecules with antioxidant properties can target the key players of inflammation. The decrease of signalling generating ROS is consequent to the inhibition of COX-2, and the antioxidant effect can be considered an effective indicator of the inhibition of this enzyme. Phenylpropanoids are widely recognised for their high prevalence in phenylalanine-derived plant metabolites. However, certain toxic mushrooms also contain notable phenylpropanoid-derived compounds. Bis-noryangonin, for example, has been isolated from the hallucinogenic mushroom *Gymnopilus spectabilis* and has undergone testing for its antioxidant properties [88]. Positive results oriented towards an in silico evaluation of the affinity of this molecule with the COX-2 receptor, which was comparable to efficient COX-2 inhibitors (−8.17 kcal/mole) [78]. These authors additionally report a good level of affinity at the COX-2 receptor for orellanine (extracted from genus *Cortinarius*), illudacetalic acid, and pulvinic acid (extracted from *Omphalotus illudens* (Schwein.) Bresinsky and Besl).

Erinacine A, a bioactive compound of *Hericium erinaceus* (Bull.) Persoon, was tested to evaluate the anti-neuroinflammatory and neuroprotective effects against neurodegenerative diseases. Lee et al. [89] observed its effects on lipopolysaccharide-induced glial cell activation and neural damage in vitro and in vivo and concluded that erinacine A pre-treatment prevented the expression of proinflammatory factors such as lipopolysaccharide-induced iNOS and NO in BV-2 cells and TNF-α in CTX TNA2 cells.

*Antrodia camphorata,* (M.Zang and C.H.Su) Sheng H.Wu, Ryvarden and T.T.Chang, was extensively studied for anti-inflammatory and anticancer properties [90,91]. Its bioactivity was dominantly attributed to the high contents of triterpenoids, benzoquinone derivatives, lignans, and polysaccharides, and its anti-inflammatory potential was mainly attributed to the direct inhibition of iNOS, COX-2, and cytokines [92].

The glucocorticoid receptors (GRs) are involved in the regulation of the inflammatory response, but their signalling pathway is also implicated in the regulation of cancer stem cells, being involved in the epithelial-to-mesenchymal transition (EMT). In fact, in GR-positive bladder cancer cells, a GR-agonist (dexamethasone) significantly reduced the expression of CD44, transcription factors, including β-catenin and its downstream target (C-MYC, Snail, and OCT-4), the rate of sphere formation, and the proportion of side populations, and induced the intracellular levels of reactive oxygen species. In contrast, GR silencing in bladder cancer cells showed the opposite effect: the enhancement of tumour growth [93]. Maruca et al. [78] reported in silico high affinity among GR and four chemicals extracted from mushrooms: bis-noryangonin (from *Gymnopilus punctifolius* (Peck) Singer), orellanine (from genus *Cortinarius*), pterulinic acid (from genus Pterula, *Omphalotus olearius* (DC.) Singer), and pulvinic acid (from *Retiboletus griseus* (Frost) Manfr. Binder and Bresinsky). Considering the affinity of the above molecules for GR receptors, it might be thought that they could possess an interesting anti-cancer activity due to the combination of their anti-inflammatory action and the ability to counter EMT.

A further receptor system that plays an important role in enabling cancer cells to evade antitumour immunity and thus has become an important therapeutic target [94] is represented by the four transmembrane receptors (A1R, A2AR, A2BR, and A3R) that belong to the G-protein-coupled receptor superfamily which bind adenosine. By binding to A1R or A3R, adenosine negatively regulates adenylyl cyclase, thereby reducing cAMP production [95], whereas by binding to A2AR or A2BR, adenosine increases cAMP production, leading to activation of protein kinase A (PKA) [96] and phospholipase C (PLC) [97]. Among these adenosine receptors, the A3R attracts attention as a therapeutic target because these receptor agonists were shown to inhibit the growth of melanoma, colon, and prostate carcinoma both in vitro and in vivo [98,99,100]. Furthermore, according to a study by Madi et al. [101], elevated expressions of adenosine A3 receptors were observed in human colon and breast carcinomas as well as lymph node metastatic tissues, in comparison to the adjacent healthy or non-neoplastic tissues., while Jafari et al. [102] demonstrated that an A3AR agonist reduces mammosphere formation in a dose-dependent manner in breast cancer stem cells by inducing G1 cell cycle arrest and apoptosis. The findings were elucidated through a western blot assay, revealing that the A3AR agonist effectively suppresses the expression of key proteins involved in cell cycle regulation and apoptosis, as well as ERK1/2 and GLI-1 proteins. Nakamura et al. [103] demonstrated that cordycepin, one of the components of *Ophiocordyceps sinensis* (Berk.) G.H.Sung, J.M.Sung, Hywel-Jones and Spatafora inhibited the proliferation of B16-BL6 mouse melanoma and Lewis lung carcinoma cells in vitro by the stimulation of adenosine A3 receptors on their cell membranes.

## 4. Direct Interaction with Networks and Receptors Involved in Cancer Stem Cells

Numerous signalling pathways become part of intertwined networks of mediators that regulate the growth of CSCs. They normally contribute to normal stem cells’ survival, proliferation, self-renewal, and differentiation properties but are abnormally activated or repressed in tumourigenesis or CSC development [104]. The complex pathways involved in regulating the behaviour of cancer stem cells (CSCs) are largely influenced by both endogenous and exogenous genetic factors, as well as microRNA activity. These pathways can result in the activation of various downstream genes, including cytokines, growth factors, apoptosis genes, antiapoptotic genes, proliferation genes, and metastasis genes, collectively contributing to the complex nature of CSCs [105]. We will limit ourselves here to summarising selected topics related to the subject areas.

### 4.1. AMPK and the PI3K/Akt/mTOR Pathway

AMPK, or AMP-dependent protein kinase, plays a pivotal role in regulating lipid and carbohydrate metabolism as well as protein synthesis. It is also known for its involvement in cancer prevention and therapy. AMPK, in fact, can promote the initiation of autophagy, and it negatively regulates the mTOR signal pathway, resulting in the inhibition of cancer proliferation; it negatively regulates COX-2, a pro-inflammatory enzyme associated with tumorigenesis; and it can induce phosphorylation of tumour suppressor p53, resulting in cell cycle arrest [106,107]. AMPK is expressed in various cancer stem cells; for example, in prostate CSCs, it maintains glucose balance [108], and in colorectal stem cells and hepatocellular carcinoma cell lines, it regulates cell proliferation and metabolism. Evidence of this is shown by the fact that its inhibition by metformin, an antidiabetic drug also used in cancer therapy, could inhibit precisely the cell proliferation and metabolism in the latter two cell lines [109,110].

The AMPK signalling pathway is a complex system that involves numerous activators, including the serine/threonine mammalian target of rapamycin (mTOR). This pathway plays a crucial role in adapting cellular anabolic activities based on the availability of various nutrients and growth factors in the microenvironment. In many types of cancer, mutations activate mTOR, which is a critical controller of metabolic homeostasis. Therefore, targeting mTOR with pharmacological and genetic tools has the potential to elucidate the oncogenic and therapeutic possibilities of modulating this pathway. Structurally and functionally, mTOR links with other proteins and serves as a core component of two distinct protein complexes, mTOR complex 1 (mTORC1) and mTOR complex 2 (mTORC2), which regulate different cellular processes [111]. mTORC1 mediates the decision of a cell to grow by integrating various growth-related inputs (protein synthesis, ribosome biogenesis, nutrient transport, lipid synthesis, and other processes in response to nutrients, growth factors and cellular energy) [112]. Its inhibition downregulates the expression of a consistent number of metabolic genes related to glycolysis, the pentose phosphate pathway, and lipid biosynthesis, resulting in a stalling of cell growth and in the release of autophagic processes to restore or maintain energy and nutrient levels. mTORC1 activation also increases aldehyde dehydrogenase 1, an effective indicator of stemness activity in colorectal CSCs [113]. Because of this, the mTORC1 effector has been identified as a target for anticancer strategies, although limited success has been achieved in this regard. Specifically, the use of rapamycin analogues (rapalogues) has not resulted in complete inhibition of mTORC1 phosphorylation and has instead led to compensatory upregulation of mTORC2-AKT activity [112,114]. mTORC2, instead, mediates actin cytoskeletal organisation, and, in cancer biology, is implicated in tumour cell motility, invasiveness, and metastasis, playing a critical role in tumour growth and survival. Gulhati et al. [115] showed how the inhibition of the mTORC2 protein results in a significant reduction of Akt phosphorylation in both rapamycin-sensitive and rapamycin-resistant colon-rectal cancers. Moreover, mTORC2 inhibitors have been shown to decrease protein synthesis, attenuate cell cycle progression, and inhibit angiogenesis in multiple cancer cell lines as well as in human cancers [116]. Numerous studies have indicated a strong correlation between the mTOR signalling pathway and the metabolism of CSCs. Specifically, low folate stress has been shown to reprogram metabolic signals by activating the mTOR signalling pathway, promoting the metastasis and tumorigenicity of lung cancer stem-like cells. These findings highlight the importance of understanding the role of mTOR signalling in CSC metabolism and suggest potential therapeutic targets for cancer treatment [117]. Moreover, the activation of mTOR promotes the survival and proliferation of breast CSCs and nasopharyngeal carcinoma stem cells [118]. The activity of mTOR is intricately linked to that of Akt, a serine/threonine kinase protein that encompasses three isoforms (Akt1, Akt2, and Akt3). Akt plays a crucial role in vital cellular functions, including cell size, cell cycle progression, regulation of glucose metabolism, genome stability, transcription, protein synthesis, and neovascularisation. By mediating cellular growth factors, Akt promotes cell survival while inhibiting apoptosis by inactivating pro-apoptotic proteins [119]. Studies have proven that the Akt signalling pathway frequently malfunctions, not only in various types of cancer, many of which are associated with its upregulation but also in cancer stem cells [120]. In CSCs, the PI3K/Akt/mTOR pathway is abnormally activated [121], and it appears to be this pathway that maintains stemness by inhibiting the MEK/ERK signalling pathway [122]. The last element not yet described here of the PI3K/Akt/mTOR pathway is PI3K. Phosphoinositide 3-kinases (PI3Ks) are a large family of intracellular signal transducers that participate in the regulation of a variety of cellular processes, including cell adhesion, cell cycle progression, cell migration, cell survival, differentiation, metabolism, proliferation, and transcription. The abnormal activation of PI3K/mTOR signalling is found in some cancers, such as non-small-cell lung cancer, breast cancer, prostate cancer, Burkitt lymphoma, oesophageal adenocarcinoma [123], and colorectal cancer [124]. Its activation is also demonstrated in the CSC population of prostate, pancreatic, and head and neck cancer, in which it promotes survival, maintenance of stemness, and tumourigenicity [125,126]. Moreover, Dubrovska et al. [125] also verified that the inhibition of the PI3K pathway by specific inhibitors led to a relative decrease in stem-like populations in prostate cancer cell lines, proving the importance of this network in the CSC viability and maintenance.

As demonstrated above, AMPK and PI3K/Akt/mTOR pathways under incorrect modulation characterise cancer stem cells and, for this reason, are considered important therapeutic targets. Hibdon et al. [127], for example, suggested that the inhibition of the PI3K/Akt/mTOR pathway could prevent proliferation and promote apoptosis in CSCs. Furthermore, it has been indicated that natural products such as quercetin, myricetin, and kaempferol, also extracted from mushrooms [128,129], could inhibit the proliferation of CSCs by inhibiting the PI3K/Akt/mTOR pathway, as shown by Li et al. [130]. Fungal extracts or molecules of fungal origin can interact with these networks by modulating these factors in multiple ways (Figure 4). Sohretoglu et al. [131], for example, demonstrated that a standardised *G. lucidum* extract, containing both polysaccharides and triterpenes, could activate AMPK and inhibit IGFR/PI3K bringing a cascading inhibition of cell proliferation and induce cell death by suppressing the mTORC2-mediated phosphorylation of Akt in human lung cancer cells (A549 and A427 cells). The extract from the mushroom *Inonotus obliquus* (Ach. ex Pers.) Pilát (Chaga mushroom) could inhibit cancer cell growth and induce autophagy by increasing AMPK phosphorylation and inhibiting the mTOR signalling pathway in terms of its downstream effectors S6 and S6K1. Moreover, inotodiol and trametenolic acid were identified as responsible for inhibiting cell proliferation without interfering with the cytotoxic effects of conventional chemotherapies [132]. Xu et al. [133] observed that oral administration and intraperitoneal injection of *Pleurotus pulmonarius* (Fr.) Quél. extract significantly inhibited tumour growth in xenograft BALB/c nude mice by triggering a marked suppression of the PI3K/AKT signalling pathway in liver cancer cells in vitro and in vivo. Moreover, they demonstrated that overexpression of the constitutively active form of AKT, Myr-AKT, abrogated this effect and inhibited proliferation and invasion showed by the *P. pulmonarius* extract. On the other hand, molecules of fungal origin antroquinonol, a ubiquinone derivative isolated from the mycelium of *A. camphorate*, perform their anti-cancer activity by modulating the AMP-activated protein kinase (AMPK). It induced apoptosis and autophagy in pancreatic cancer cells [134], inhibition of the cellular growth of liver and lung cancer cells [135], blockades cellular protein synthesis through inhibition of protein phosphorylation [136], and it inhibited migration and invasion of breast cancer through suppressing expressions of matrix metalloproteinase (MMPs) and epithelial-mesenchymal transition (EMT) genes [91]. Hispolon, from *Phellinus lonicerinus* Quel., suggests positive effects against the migration and invasion of human nasopharyngeal carcinoma by inhibiting uPA via the modulation of the Akt signalling pathway [137].

### 4.2. The Epithelial-Mesenchymal Transition (EMT)

The EMT is a process by which epithelial cells undergo remarkable morphological and physiological changes in which they break down cell–cell and cell–extracellular matrix connections and gain migratory and invasive properties. Cells undergoing EMT can acquire stem cell-like characteristics, which indicates an interesting conjunction between EMT and stem cells [138,139,140]. Several studies also highlighted the link between EMT and cancer stem cells: Mani et al. [141] found that EMT induction in human mammary epithelial cells could lead to the acquisition of mesenchymal morphology and the expression of mesenchymal markers, which increased the subpopulation with stem cell properties with tumour-initiating ability. Similar results were found by Morel et al. [142]. Moreover, Gupta et al. [133] conducted a study demonstrating the induction of epithelial-to-mesenchymal transition (EMT) in transformed HMLER breast cancer cells. This led to an increase in the population of CD44+/high/CD24−/low cells, resulting in a significantly greater mammosphere-forming ability and increased drug resistance related to the biology of cancer stem cells (CSCs). Additionally, research has shown that cancer-associated fibroblast-induced EMT in prostate carcinoma cells resulted in the overexpression of stem cell markers and the formation of spheres and self-renewal [143]. One event that allows the attached epithelial cells to transpose to mesenchymal status is the inhibition of E-cadherin. The glycogen synthase kinase-3 beta (GSK-3 beta) regulates the transcription of E-cadherin through phosphorylation of Snail and β-catenin to trigger proteasomal degradation [144] (Figure 5). Šeklić et al. [145] demonstrated that methanolic extract of *Phellinus linteus* (Berk. and M.A.Curtis) and *L. edodes* increased the level of E-cadherin and reduced the nuclear β-catenin in HCT-116 and SW-480 cells, bringing, as a consequence, the suppression of promigratory markers N-cadherin and vimentin. Similar results were obtained by de Camargo et al. that showed how an extract from *G. lucidum* could successfully aid the prevention of the EMT process in SCC-9 cells and possibly at tongue carcinoma in situ, downregulating TWIST, AXL, vimentin, and N-cadherin (molecular markers of EMT), reinforcing the predominately epithelial phenotype induced by polysaccharide treatment [146].

### 4.3. Matrix Metalloproteinase (MMPs) in EMT

MMPs, a family of extracellular zinc-dependent endopeptidases, play an essential role in EMT; they are engaged in stem cell niches during development by rearranging the extracellular matrix for tissue remodelling and organ development. They specifically modulate signalling pathways through proteolytic interaction with multiple substrates. MMPs have been identified as significant contributors to tissue invasion during both developmental processes and tumour progression. MMPs are known to facilitate cellular migration and tissue invasion through the cleavage of extracellular matrix (ECM) and basement membrane components, thereby creating a pathway for invasive tissues or cancer cells to navigate through the interstitium. This widely considered mechanism highlights the crucial role of MMPs in promoting tissue invasion and underscores the importance of understanding their function in these processes [147]. Kumar et al. [148] demonstrated that such prometastatic molecules, particularly MMP-9, enhance stem cell features to promote tumorigenesis and metastasis in triple-negative breast cancer. In a study conducted by Inoue et al. [149], it was suggested that the invasive potential of cancer stem cells is largely dependent on the enzymatic activity of MMP-13. The research revealed that the expression of MMP-13 was notably elevated in tumoursphere-forming cells derived from U251 and primary human glioma cells. Furthermore, the knockdown of MMP-13 expression by shRNA was found to suppress the migration and invasion of these cells. Given the substantial experimental and clinical evidence linking MMPs to tumour progression and poor prognosis, these enzymes represent crucial therapeutic targets. Once again, medicinal mushroom extracts reveal their potential: Šeklić et al. [145] demonstrated that methanolic extract of *P. linteus* and *L. edodes* induced the suppression of cell invasion, this time by reducing the level of MMP-9, and Liu et al. [150] indicated the downregulation activity of a commercial product containing a mixture of spores and fruiting bodies of *G. lucidum* towards the genes MMP-10, 12, and 13 in treated mice. However, other fungal extracts exhibited MMP-inhibitory activity, for example, *Dictyophora indusiata* (Vent.) Desv. extract against MMP-2 [151], and an extract of the mycelium of *Tricholoma matsutake* (S.Ito and Imai) Singer against MMP-1 [152] (Figure 5). Hseu et al. [153] also demonstrate the in vitro anti-metastatic properties of a fermented culture broth of *A. camphorata* in a line of human colon cancer cells (SW620^claudin−1+^). It significantly decreased the protein levels of MMP-9 and MMP-2, suggesting that the fermented culture broth possesses potentially anti-migratory and anti-invasive properties.

### 4.4. KRAS and IMPs, Two Cornerstones of EMT

The process of epithelial-mesenchymal transition (EMT) is triggered by microenvironmental cues and controlled by a network of EMT-inducing transcription factors (EMT-TFs) that collaborate with epigenetic regulators to modulate the expression of key proteins responsible for maintaining cell polarity, cell–cell communication, cytoskeletal organisation, and extracellular matrix (ECM) degradation [154]. One of the genes involved in cellular transition is KRAS, which encodes a small membrane protein that becomes active upon binding to GTP. This activation subsequently triggers several downstream biochemical pathways, leading to cell growth, differentiation, and survival. KRAS also plays a role in recruiting and activating proteins necessary for the propagation of growth factors and other cell signallings receptors such as c-Raf and PI 3-kinase [155]. The oncogene in question is known to undergo frequent mutations in human cancer cells. It is particularly prevalent in pancreatic adenocarcinomas, colorectal cancers, and lung cancers [156]. The mutation of KRAS results in its constant binding to GTP in the “on” state, leading to the continuous activation of downstream biochemical effectors that can potentially cause tumour formation [157]. KRAS mutation also activates CSCs, contributing to colorectal tumourigenesis and metastasis in colorectal cancer cells. Another cornerstone of EMT consists of the IMPs proteins. They are highly conserved oncofetal RNA-binding proteins that regulate RNA processing at several levels, including localisation, translation, and stability. During embryogenesis, three mammalian IMP paralogs, namely IMP1, IMP2, and IMP3, are expressed in most organs. These paralogs are believed to have a crucial function in cell migration, metabolism, and stem cell renewal. In addition, studies have shown that while IMP2 expression is maintained in various adult mouse organs, IMP1 and IMP3 are either absent or expressed at low levels in most tissues post-birth [158]. However, all three paralogs appear to resume their physiological functions in a broad range of tumour types, where their expression often correlates with poor prognosis [159,160,161]. Tessier et al. [162] verified that transgenic mice engineered to express IMP1 in mammary epithelial cells developed mammary tumours in up to 95% of cases, some of which formed metastases. IMPs also seem to be correlated with EMT (and thus also of the presence of CSC): You et al. [163] demonstrated that epithelial-mesenchymal transition in colorectal carcinoma cells was mediated by DEK/IMP3; in fact, once this network was silenced, the level of E-cadherin was enhanced and the expression of vimentin and MMP-9 were apparently downregulated, influencing the invasion of colorectal carcinoma cells negatively. Hamilton et al. [164], given the fact that increased IMP expression correlates with enhanced metastasis and poor prognosis in colorectal cancer, observed how IMP1 overexpression modulates tumour dissemination into the blood, promotes colonosphere growth from single tumour cells, and enriches the population of CD24+CD44+ expressing cells, suggesting that IMP1 modulates tumour growth and dissemination of tumour cells into the blood. Yaqoob et al. [165] isolated three compounds from the fungi *Albatrellus flettii* Morse ex Pouzar that exhibited cytotoxic activity towards colon cancer cells. These three known compounds were grifolin, neogrifolin, and confluentin. The findings of the study indicate that confluentin exhibits the ability to initiate apoptosis and halt the cell cycle at the G2/M phase, in addition to its role in suppressing KRAS expression in colon cancer cells. Additionally, neogrifolin, grifolin, and confluentin itself have demonstrated this suppression effect. Furthermore, confluentin has been observed to specifically inhibit the physical interaction between the oncogenic insulin-like growth factor 2 mRNA-binding protein 1 (IMP1) and KRAS RNA.

### 4.5. p53 Expression in EMT

The mechanisms described so far relate EMT to CSCs, but they are not the only ones. The protein p53, known as a tumour suppressor, plays a crucial role in regulating the balance between self-renewal and differentiation of stem cells necessary for the maintenance and proper development of tissue homeostasis. In the context of malignancies, the inactivation of p53 disrupts this equilibrium, leading to aberrant pluripotency and somatic cell reprogramming [166]. Chang et al. [167] shed light on the importance of this molecule in the regulation of EMT and EMT-associated stem cell properties. They demonstrate that a loss of p53 synthesis in mammary epithelial cells could decrease the expression of miR-200c and activate the EMT program, accompanied by an increased mammary stem cell population. Research findings indicate that the reversal of a particular process in normal cells is feasible. Specifically, when TGF-β was removed from MCF12A cells that were pre-treated with TGF-β for 14 days, the p53 expression level was promptly restored within four days. This restoration was accompanied by a significant increase in the expression of epithelial marker E-cadherin and a decrease in the CD24−CD44+ stem cell population. These results suggest that the process could potentially be reversed in normal cells. Several studies demonstrated that fungi metabolites could interact with this network and could function in cancer prevention and therapy. For example, Jedinak et al. [168] demonstrated that the dietary mushroom *P. ostreatus* specifically inhibits the growth of colon and breast cancer cells without significant effect on normal cells. The methanolic extract of this mushroom caused cell cycle arrest at the G0/G1 phase, induced the expression of p21, p53, p27, and p19, and downregulated the expression of CDK4, CDK6, Ki67, E2F transcription factor 1 (E2F1), transcription factor Dp-1 (TFDP1), and proliferating cell nuclear antigen (PCNA) genes in breast cancer MCF-7 cells. The immunomodulatory protein rLZ-8, extracted from *G. lucidum*, expressed a similar activity against lung cancer cells A549 both in vitro and in vivo: it arrested the cell cycle in the G1 phase and showed a growth arrest effect in a p53-dependent way [169]. Orellanine (a metabolite of the fungi of the genus *Cortinarius*) is a nephrotoxic agent specifically targeting proximal tubular epithelial cells, leaving other organs unaffected. Interestingly, its toxicity also extends to renal carcinoma clear cells both in vitro and in vivo. Buvall et al. [170] showed that orellanine significantly disrupts mitochondrial function, lowering mitochondrial respiration and glycolysis following exposure. It also acts as a DNA-damaging agent, as demonstrated by the increased phosphorylation of p53 at serine 15 (a site commonly phosphorylated following DNA damage). Bringing the stabilisation of p53 seems to inhibit the interaction between p53 and its negative regulator MDM2. According to the authors, orellanine toxicity is not only due to DNA damage but also to the promotion of ROS generation. It is known that CSCs can dynamically modify their metabolic state to favour glycolysis or oxidative metabolism [171]. Therefore, this molecule targeting CSCs metabolism may provide new and effective methods for the treatment of tumours. It has also been demonstrated that a fraction of a purified extract of *Pleurotus highking* (PEF-III) possesses strong anticancer activity via induction of apoptosis by alteration of the balance of apoptosis-related genes [172]. It upregulated p53 and its target gene, Bax, and downregulated the expression of the antiapoptotic gene Bcl-2. According to Lenzi et al. [173], the ethanolic extract of *Meripilus giganteus* (Pers.) Karst. could perform the same action by increasing the Bax/Bcl2 ratio in the HL-60 cell line.

### 4.6. The Receptor Tyrosine Kinases (RTKs) Targeted by Medicinal Mushroom

In addition to the ones described above, several signals are known to be associated with the stemness of CSCs and, in particular, with the epithelial-mesenchymal transition. Receptor tyrosine kinases (RTKs) hold a crucial position in shaping various biological processes, including embryogenesis, organogenesis, and tissue regeneration.

c-Met and epidermal growth factor receptor (EGFR) are among the most well-studied RTKs.

c-Met, also called tyrosine-protein kinase c-Met or hepatocyte growth factor receptor (HGFR), plays a role in the epithelial-mesenchymal transition [174] by activating WNT-β-catenin signalling cascade to promote stemness and invasion [175]. Moreover, the c-Met ligand increases β-catenin transcriptional activity through phosphorylation to regulate migration and invasion [176,177]. Studies using mouse models have revealed that constitutive activation of c-Met in basal-like breast cancer can impede the differentiation of mammary luminal progenitor cells and induce traits associated with stem cells [178]. In prostate and pancreatic cancer, both HGF and MET genes are reported to be preferentially expressed in stem-like tumour cells [179,180]. In glioblastoma, MET activation promotes stemness and induces the invasive phenotype [181]. In xenograft models, cMet inhibitors (anti-HGF, anti-MET, and MET-targeted small molecule inhibitors) decrease tumour progression and the expression of stem markers such as CD133, Sox2, and Nanog [182]. Thus, these studies collectively support the roles of MET signalling in CSC maintenance. According to related literature, both mushroom alcoholic extracts and pure molecules derived from fungi could interfere with the c-Met pathway. On the one hand, Li et al. [183] reported a significant decrease in the mRNA expression of Fra-1, c-Met, and vimentin in HCT116 cells following treatment with ethanol extracts of sporoderm-broken spores of G. lucidum, which are primarily composed of triterpenoids. The observed downregulation was particularly pronounced at higher doses of the extract. On the other, Lu et al. [184] showed that verticillin A, a diketopiperazine compound isolated from *Amanita flavorubescens* G.F. Atk., could effectively inhibit HGF-induced c-Met phosphorylation and repressed the expression of the total c-Met protein in in vitro AGS and HeLa cells. Moreover, by impairing c-Met phosphorylation, verticillin A could suppress c-Met downstream FAK/Src signalling pathways in vitro. One year later, the same research group observed that verticillin A could significantly inhibit the migration and invasion of murine colon cancer cells by targeting c-Met and inhibiting Ras/Raf/mitogen-activated extracellular signal-regulated kinase (MEK)/ERK signalling pathways in in vivo [185]. Another approach attempted is the in silico theoretical approach, in which fungal molecules are studied in relation to their probable efficacy against specific pathways. Maruca et al. [78] showed that several mycochemicals exhibit a high level of affinity for cMet (reported as free energy being in brackets): Blennin C (10.16 kcal/mole) from *Russula Sanguinaria* (Bull.) Fr., Hericenol B (9.52 kcal/mole) and Hericenone H (8.82 kcal/mole) from *H. erinaceus*, 1-O-Acetyl-3-epi-illudol (8.97 kcal/mole) from *Clitocybe candicans* (Pers.) P. Kumm., (R)-Torosachrysone (8.96 kcal/mole) from Genus *Cortinarius*, and illudinine (8.89 kcal/mole) from *H. erinaceus*.

The transmembrane glycoprotein known as the epidermal growth factor receptor (EGFR) is a significant member of the erbB family of tyrosine kinase receptors. Recently, Abhold et al. [186] conducted a study to investigate the role of EGFR in regulating “stemness” in head and neck squamous cell carcinoma cells. The study’s findings shed light on the potential implications of EGFR in the progression and treatment of this disease. These authors activated EGFR by the addition of EGF ligand or ectopic expression of EGFR in two established cancer cell lines (UMSCC-22B and HN-1). This activation resulted in the induction of CSC markers, such as CD44, BMI-1, Oct-4, NANOG, CXCR4, and SDF-1. The activation of the epidermal growth factor receptor (EGFR) has been shown to enhance the formation of tumourspheres, a defining trait of cancer stem cells (CSCs). Conversely, treatment with the EGFR kinase inhibitor, Gefitinib, results in a decrease in expression of the aforementioned genes and a loss of tumoursphere-forming ability. The suppression of tumoursphere formation is a crucial factor in identifying selective inhibitors of CSCs through high-throughput screening [169]. As introduced before, a pharmacological approach consisting of tyrosine kinase inhibitors (TKIs) has been developed to target EGFR, but, unfortunately, a high incidence of resistance phenomena is reported. For this reason, there is an increased necessity to identify agents that may be combined with these therapies to provide a sustained response for cancer patients. Suárez-Arroyo et al. [187] evaluated the therapeutic potential of the *G. lucidum* extract ReishiMax GLp™ (GLE) alone and in association with Erlotinib in pharmaco-sensitive (SUM-149) and pharmaco-resistant (rSUM-149) breast carcinoma cell lines. The research highlights the ability of GLE to synergise with Erlotinib to sensitise cancer cells to drug treatment and overcome intrinsic and developed Erlotinib resistance. This combination decreases SUM-149 cell viability, proliferation, migration, and invasion. Moreover, the mushroom extract increases Erlotinib sensitivity by inactivating AKT and ERK signalling pathways in vitro and in vivo models. Probably, Lin et al. [188] identified a molecule of *G. lucidum* extract responsible for the described activity, the LZ-8 protein. They demonstrated that its recombinant version, rLZ-8, induced cycle arrest and apoptosis by downregulating the expression of EGFR and inhibiting EGFR downstream effectors, AKT and ERK1/2 in lung cancer cells. In parallel, He et al. extracted a novel protein from *T. versicolor,* called musarin, with the peculiarity of attenuating the colorectal cancer stem cells proliferation, in vitro, by down-regulating multiple signalling pathways, including the EGFR-Ras signalling pathway. He et al. [189] also showed that the oral ingestion of musarin inhibits human colorectal tumour development in SCID/NOD mice with no observed side effects. Moreover, other metabolites from the mushroom *H. erinaceus* also showed a possible action against EGFR: Hericenone H and Hericenol B exhibited an affinity for EGFR (−8.48 kcal/mole and 8.41 kcal/mole, respectively), hypothesising a possible inhibitory activity [78].

## 5. Extraction Approaches for Mycochemicals

In the studies reviewed, the molecules or fractions utilised were extracted from fungal matrices such as the fruiting body or mycelium. It is crucial to note that obtaining extracts with high therapeutic value requires the use of an optimized and standardised extraction method, which ensures the highest possible metabolite content and consistent concentration and quality of metabolites across repeated batches. The complexity of the chemical characteristics of the metabolite of interest, particularly when molecules of different chemical classes such as glucans, polyphenols, and terpenes are considered, necessitates the utilisation of varying extraction processes.

The extraction approaches to obtain the best yield of β-glucans from fungi have long been optimised and consolidated, and they include water, alkali, and acid extraction. Alkali and acid extraction could change the viscosity of β-glucan, while the process of hot water extraction has been found to achieve a higher extraction rate of water-soluble β-glucan. As reported by Yang and Huang [190], the extraction process on properly freeze-dried and pulverised fungal matrices is characterised by sequenced steps of heating (in boiling water) and freezing (at −20 °C with 1–4 volume of absolute ethanol) to obtain the highest yield of precipitated polysaccharide fractions. Further passages (e.g., filtration on dialysis membrane or centrifugation) are then necessary to separate impurities to obtain purified β-glucans.

However, as regards the extraction of other molecules with possible antitumoural activity, the approaches on fungi are similar to those adopted on plants or different natural matrices. The extraction approaches are generally optimised according to the type of natural matrix to be extracted (e.g., organ, consistency, nature, etc.) and the properties of the molecules to be extracted, e.g., concerning their polarity, even before their biological activity. Depending on these parameters, the methods to be used may require different types of solvents or their mixtures, and different instrumental approaches, from those that are traditional, such as maceration, distillation, and Soxhlet, to those closer to the most current needs of sustainability, such as supercritical fluid extraction, pressurised fluid extraction, ultrasound-assisted extraction, microwave-assisted extraction, etc. These latter extraction methods have their sustainability goal in matching qualitative and quantitative yield of bioactive molecules, with a low environmental impact process, mainly through the reduction of organic solvent use and the full exploitation of the raw material, avoiding any waste, therefore also using by-products as a source of bioactive compounds [191].

## 6. Conclusions

The advances in stem cell biology have facilitated research in multiple aspects of CSC characteristics. Recent studies have successfully identified stem cell sub-components in human malignancies. Moreover, significant progress has been made in understanding the key signalling pathways that are involved in stem cell self-renewal and differentiation. These findings have proven instrumental in facilitating the selective targeting of cancer stem cells (CSCs). Clinical trials that use mushroom extracts or mycochemicals, although not specifically targeting CSCs, also take their biological mechanisms into account [192]. Metastatic processes, drug resistance, and clonogenic ability represent the functional expression of the presence and development of CSCs, and mushroom derivatives are active in these contests. In fact, as reported in this review, mushrooms are an abundant source of mycochemicals that may intervene in several pathways involved in the induction of embryonic characteristics specific to CSCs, either alone or in synergy with conventional drug therapies. Considering the above, but also the fact that most of the fungal species inhabiting the Earth have yet to be discovered, the intensification of mycology and mycology research applied to cancer treatment and prevention could make the fungus a strategic source for the future.

## Figures and Tables

**Figure 2 toxins-15-00360-f002:**
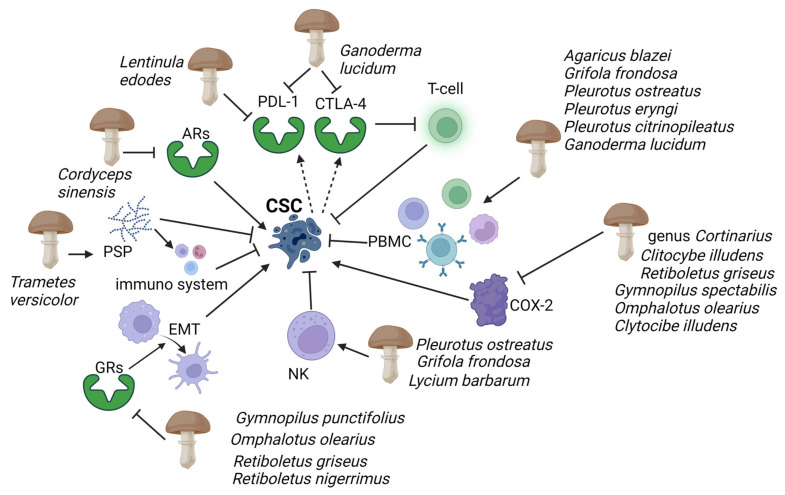
Evidence has highlighted mushroom extracts as potential toxicological agents against CSC mediated by the immune system. Polysaccharide peptides (PSP) from *Trametes versicolor* appear to work as a biological response modifier, enhancing the macrophages and T-lymphocytes’ activity. Adenosine receptors (ARs), which play an essential role in enabling cancer cells to evade antitumour immunity, are inhibited by *Ophiocordiceps sinensis* extracts. Programmed cell death ligand 1 (PD-L1) and cytotoxic T-lymphocyte-associated protein 4 (CTLA-4) are activated by CSCs (dashed arrows), causing immune cell dysfunction; *G. lucidum* thwarts this action by inhibiting PD-L1 and CTLA-4. Immunosuppressive action of human peripheral mononuclear cells (PBMC) and natural killer (NK), and COX-2 inducing cancer stem cell (CSC)-like activity on CSCs are enhanced and inhibited respectively by several mushrooms (see figure). *Gymnopilus punctifolius*, *Omphalotus olearius*, *Retiboletus griseus,* and *Retiboletus nigerrimus* extracts hold competitive antagonists of glucocorticoid receptors (GRs) which, through activation of epidermal mesenchymal transition (EMT), contribute to increases CSCs population. Thin arrows represent activation and T shape segment inhibition. The image was created in BioRender.com (Agreement number: AW253P4DHI).

**Figure 3 toxins-15-00360-f003:**
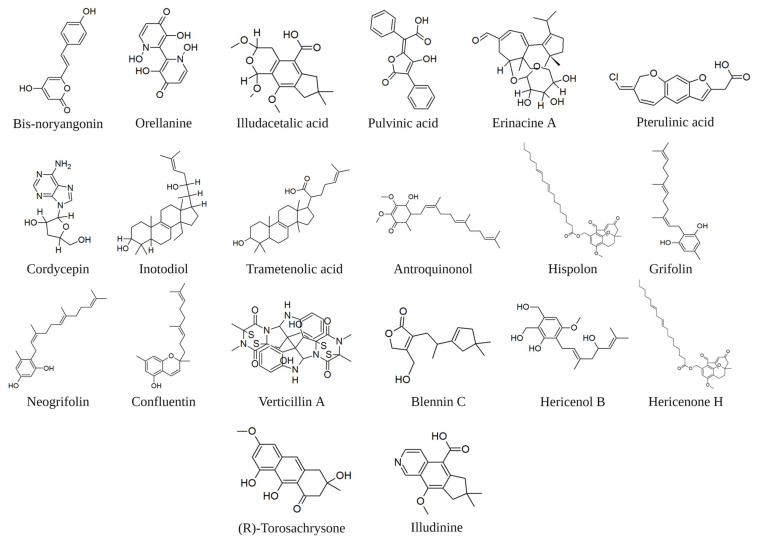
Examples of molecules of fungal origin cited in the text. The image was created in BioRender.com (Agreement number: QV254UDWM6).

**Figure 4 toxins-15-00360-f004:**
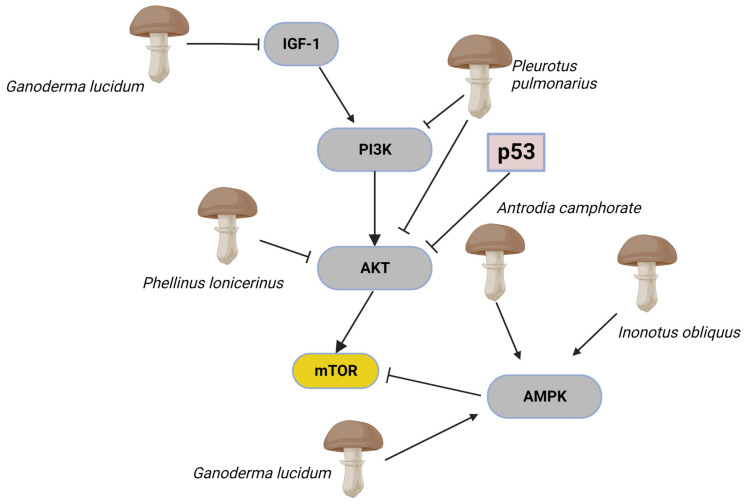
The complex mTOR/Akt signalling pathways are frequently co-activated and altered during oncogenesis. In cells, mTOR activity is controlled by positive and negative upstream regulators. Positive regulators include growth factors and their receptors, such as insulin-like growth factor-1 (IGF-1) and its cognate receptor, members of the human epidermal growth factor receptor, which transmit signals to mTOR through the PI3K-Akt. Inhibition of PI3k and modulation of Akt affects mTOR signalling activity. The guardian of metabolism and mitochondrial homeostasis, AMP-activated protein kinase (AMPK) can also operate as an mTOR inhibitor. This section of the mTOR/Akt signalling pathways has shown to be inhibited by mushroom extracts. Extracts from *Pleurotus pulmonarius* inhibit PI3K and AKT. *Antrodia camphorate*, *Inonotus obliquus*, and *Ganoderma lucidum* are inhibitors of AMPK. The image was created in BioRender.com (Agreement number: EC253P43E6).

**Figure 5 toxins-15-00360-f005:**
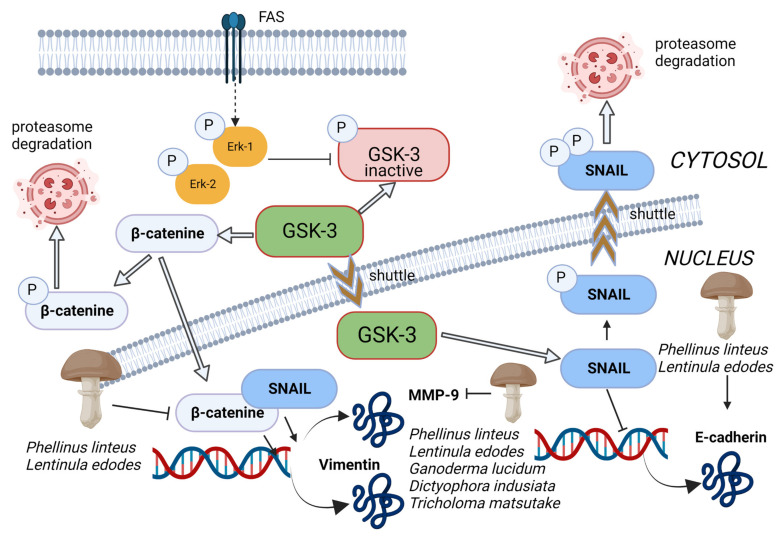
The glycogen synthase kinase-3 beta (GSK-3 beta) regulates the transcription of E-cadherin through phosphorylation of Snail and β-catenin to trigger proteasomal degradation. Inhibition of GSK-3b activity prevents the phosphorylation and subsequent degradation of β-catenin and Snail, which accumulate in the nucleus and induce EMT by inhibiting E-cadherin expression and induction of MMP9 and vimentin expression. Methanolic extract of *Phellinus linteus* and *Lentinula edodes* increased the level of E-cadherin and reduced the nuclear β-catenin in cancer cells, bringing, as a consequence, the suppression of mesenchymal and promigratory markers N-cadherin and vimentin. Moreover, *Pellinus linteus, Lentinula edodes, Ganoderma lucidum, Dictyophora indusiata,* and *Tricholoma matsutake* extracts exhibited MMP inhibitory activity. Thin arrows represent activation and T shape segment inhibition. The image was created in BioRender.com (Agreement number: TP253P4LDC).

## Data Availability

Data sharing not applicable.

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
