# Peer review of "Mycochemicals against Cancer Stem Cells"

_toxins, 2023, doi:10.3390/toxins15060360_

Round 1

Reviewer 1 Report

In this review, the authors discuss several pieces of evidence of the association between β-glucans and small mycochemicals in modulating biological mechanisms which are proven to be involved with CSCs development. Research is very important. However, there are some problems in the writing of the article that need to be improved.

In introduction: The review literature retrieval media, key words, etc. should be given. And clarify whether there are similar review articles before and the relationship with this article.

The structure of the compound needs to be redrawn, which is too vague.

The dot after the title should be removed.

It is recommended that the author further refine the content of the article and add schematic diagrams to increase readability.

I would like to know if there is any progress in the clinical application of mushroom extract that can increase the author's interest in reading, such as the following article (Front Pharmacol

. 2020 Nov 11;11:580656. doi: 10.3389/fphar.2020.580656.).

The literature in the past two years is too small and needs to be increased.

Author Response

Dear Reviewer,

we appreciate your valuable feedback that has helped us enhance our review. We have addressed your suggestions for improvement in a detailed manner:

In introduction: The review literature retrieval media, key words, etc. should be given. And clarify whether there are similar review articles before and the relationship with this article.

Reply 1: The introduction was amended as requested (lines 66-79).

The structure of the compound needs to be redrawn, which is too vague.

Reply 2: β-glucans are very complex macromolecules due to their linear chain length and branching. We have reported the main constituent monomer of the polymer for illustrative purposes. However, we have modified the caption (also adding the molecular weight of the β-glucans considered) so as to make it clear to the reader that what is shown in the figure is only the monomeric component of a more complex molecule.

The dot after the title should be removed.

Reply 3: The text was amended as requested

It is recommended that the author further refine the content of the article and add schematic diagrams to increase readability.

Reply 4: The content of the manuscript was refined as requested. Regarding the suggestion to include diagrams, we have considered the intricate nature of the biological networks involved and provided schematisation of the main ones. Moreover, given the potential for fungal-derived molecules to exert multitarget effects on various networks, we would have had to depict multiple networks for the same metabolite, which would have significantly increased the complexity of the diagrams.

I would like to know if there is any progress in the clinical application of mushroom extract that can increase the author's interest in reading, such as the following article (Front Pharmacol. 2020 Nov 11;11:580656. doi: 10.3389/fphar.2020.580656.).

Reply 5: We agree with R1 that clinical trials using fungal extracts in the fight against cancer are very important in the therapy and prevention of neoplastic diseases, which is why we have included the suggested article within the bibliography of the manuscript. Nevertheless, this review aims to be a tool to direct research on medicinal fungi and their metabolites toward specific pathways that regulate CSC formation and development. Including clinical trial information may not devote enough space to the importance of this hospital-based research by being too reductive and unable to address the topic adequately. For this reason, we aim to study the topic in more detail in the future.

The literature in the past two years is too small and needs to be increased.

Reply 6: Please note that our research indicates that all relevant articles published within the past two years have already been reviewed regarding fungal matrices and cancer stem cells. However, we have included additional literature references that analyse the impact of fungal extracts on cancer cells using specific pathways related to CSC biology.

Best regards,

Reviewer 2 Report

The present paper is interesting and well-written. However, I missed information about sample/extract preparation. Methods used for the extraction of biologically active compounds have a high impact on the final activity of the extracts. In my opinion, it would be valuable to add a section about methods used for the extraction of compounds with antitumoural properties. Methods, solvents, and purification possibilities can be discussed.

Author Response

Dear Reviewer,

We appreciate your input and share your perspective. This is why we have included a brief analysis of the extraction methodologies used to obtain pure extracts/molecules in the relevant studies. While a more detailed examination of the extractive processes would have made the topic longer and required more time, we wanted to emphasise the importance of distinguishing between extractive methods for glucans and other small molecules. We are also highlighting the potential of using environmentally friendly and sustainable green methods.

Best regards,